# A comparative analysis of infection and mortality in reassessing africa's COVID-19 dynamic using time-varying tests

Stéphane Luchini [1] ✉, Constantin Pfauwadel[2], Patrick A. Pintus[1,3], Michael Schwarzinger[4,5] & Miriam Teschl[1,6]

## Abstract

**Background** It is commonly believed that Africa largely evaded the worst of the COVID-19 pandemic, with fewer cases than other continents. However, regional comparisons that ignore differences in testing intensity may misrepresent dynamics. Studying the spread and case-fatality relationship during COVID-19 across WHO regions requires explicitly adjusting for time-varying test volumes.

**Methods** We build a weekly panel dataset spanning May 2020 to December 2021 for the WHO regions: Africa, Eastern Mediterranean, South-East Asia, the Americas, Western Pacific, and Europe. Data on tests, confirmed cases, and COVID-19-attributed deaths were sourced from Our World in Data. We apply a novel metric that corrects for fluctuating test volumes to quantify week-to-week acceleration in infections and in mortality. We then compare the frequency, magnitude, and timing of these acceleration episodes across regions.

**Results** Accounting for testing dynamics, we show that Africa exhibits multiple infection-acceleration episodes whose magnitude and frequency match those in other regions. Mortality accelerations in Africa closely follow infection surges, with an average lag of ten weeks. A positive correlation between infection acceleration in Africa and the Americas further indicates synchrony. These findings hold when using a larger secondary dataset of 140 countries.

**Conclusions** Contrary to prevailing assumptions, Africa was not spared from the pandemic's severe dynamics. Infection surges were on par with those elsewhere and were followed by mortality accelerations. These results underscore that accounting for testing variability is essential to accurately assess pandemic progression, and they highlight the urgent need to strengthen surveillance and healthcare capacity across all regions.

## Plain language summary

It is commonly believed that Africa was less affected by COVID-19 than other parts of the world because it reported fewer cases. However, differences in how much testing was done can make such comparisons misleading. In this study, we used a new measure that accounts for changes in testing levels to examine how quickly infections and deaths increased over time. We analyzed weekly data from May 2020 to December 2021 across World Health Organization regions. Our findings show that Africa experienced several infection surges similar in size and frequency to those in other regions, followed about ten weeks later by increases in deaths. Considering testing differences in surveillance and healthcare systems is crucial for accurately assessing the pandemic's course worldwide.

The relatively low number of reported COVID-19 cases and deaths in Africa has prompted debates about whether the continent was spared the worst of the pandemic, a phenomenon described by some as the African "puzzle"[1,2] or "paradox"[3]. Early media reports and research articles speculated that Africa's younger population, lower population density in rural areas, and prior experience with infectious diseases and their pharmaceutical treatments might have mitigated the severe impacts observed in other regions.

However, emerging evidence from seroprevalence studies indicates that far more individuals in Africa were exposed to SARS-CoV-2 than is reflected in official surveillance data, especially during the pandemic's first 2 years. For example, the ratio of seroprevalence to confirmed cases has been estimated to be as high as 100:1[4]. This gap between seroprevalence estimates and reported cases grew as the pandemic continued[5], suggesting substantial under-reporting in surveillance data. For example, while[6] mention low testing rates as a likely source of under-reporting in their discussion of the

[1]Aix-Marseille University, CNRS, Aix-Marseille School of Economics, Marseille, France. [2]Paris School of Economics, Paris, France. [3]Institut de Neurosciences de la Timone, Marseille, France. [4]Department of methodology and innovation in prevention, Bordeaux University Hospital, Bordeaux, France. [5]UMR 1219 Bordeaux Population Health, INSERM, Bordeaux, France. [6]EHESS, Paris, France. ✉e-mail: stephane.luchini@univ-amu.fr

literature, they do not incorporate test data in their analysis of the 12 most affected countries with the highest cumulative COVID-19 deaths.

In this context, our objective is to investigate whether findings from surveillance systems can be corrected by accounting for the time-varying nature of testing. By adjusting for variations in testing over time, we aim to bridge the gap between the low number of reported cases and the high infection estimates from seroprevalence studies, providing a more accurate assessment of the true spread and impact of COVID-19 in Africa. We address this aim by first constructing a comprehensive weekly database of cumulative COVID-19 tests, confirmed cases, and deaths across the six WHO regions, using data originally extracted from the "Our World in Data" website. The final database includes 89 countries over the period from 2020 to 2021, with cumulative tests, cases, and deaths increasing monotonically on a weekly basis. To measure the impact of test variability on case detection and mortality, we use acceleration indices from ref. 7 within each WHO region (see also ref. 8 for a similar analysis on Italian data). Acceleration indices applied to test and case data serve as test-adjusted reproduction numbers, allowing us to analyze the dynamics of infection under varying testing conditions, as shown in ref. 9. Similarly, applying acceleration indices to cases and deaths enables us to study mortality dynamics by calculating the elasticity of cumulative deaths relative to cumulative cases - effectively analyzing nonlinear shifts in case-fatality rates. We then compare the COVID-19 dynamics of infection and mortality across the different WHO regions. In this study, we hypothesize that overlooking the scarcity and variability of testing in African countries may have hindered accurate measurement of COVID-19 infection and mortality dynamics. We then assess whether adjusting for testing intensity can substantially mitigate—or even resolve—the discrepancy commonly referred to as the African "paradox".

Accounting for testing dynamics, we show that Africa exhibits multiple infection-acceleration episodes whose magnitude and frequency match those in other regions. Mortality accelerations in Africa closely follow infection surges, with an average lag of 10 weeks. A positive correlation between infection acceleration in Africa and the Americas further indicates synchrony. These findings hold when using a larger secondary dataset of 140 countries. Contrary to prevailing assumptions, Africa was not spared from the pandemic's severe dynamics. Infection surges were on par with those elsewhere and were followed by mortality accelerations. These results underscore that accounting for testing variability is essential to accurately assess pandemic progression, and they highlight the urgent need to strengthen surveillance and healthcare capacity across all regions.

## Methods
### Data on COVID-19 tests, positive cases, and deaths
Weekly data on cumulative COVID-19 cases and deaths were sourced from the "Our World in Data" website (see ref. 10). These data were provided as daily entries, where the same observation was repeated for seven consecutive days to represent each day of the week. We chose to use weekly data on cumulative cases and deaths, rather than the daily data available from the Center for Systems Science and Engineering (CSSE) at Johns Hopkins University, due to inconsistencies we encountered in the Johns Hopkins dataset. Specifically, several countries had non-monotonic cumulative case and death counts over time, which created inaccuracies. While the "Our World in Data" dataset also had a few minor inconsistencies of this type (affecting only five observations), we corrected these issues by ensuring that whenever a decrease in the cumulative number of cases or deaths occurred from one week to the next in the raw data, the weekly number of cases for the current week was replaced with the value from the previous week.

The data on the cumulative number of COVID-19 tests performed, which is crucial to our analysis, also came from the "Our World in Data" website. This international database provides open-access, harmonized, and uniform data on testing, although it is only partially documented. A significant challenge we faced was the limited availability of test data. Not all countries were represented in the dataset, and for many countries, data were

not available for every date, possibly due to tests being reported on a weekly basis or for reasons unknown to us.

Because our acceleration-index methodology critically relies on the coherence of tests, cases and deaths both over time and relative to one another, generic case–only cleaning procedures can leave undetected inconsistencies. We therefore undertook a detailed harmonization process. To ensure consistency with the case and death data, we transformed the daily test data into a weekly format. Specifically, we grouped the data by country and week (where a week is defined as starting on Sunday). For each week, we selected the cumulative number of daily tests available each Sunday as the cumulative number of weekly tests for that week. When data was missing for some days of the week, we selected the smallest cumulative number of daily tests (typically the value at the beginning of the week) and repeated that value for all days within that week until the next Sunday. This approach aligned the cumulative test data with the weekly format of the case and death data, and allowed us to include countries that reported only weekly test data. Additionally, to avoid introducing artificial variations, we retained the previous cumulative case and test values when the cumulative number of tests remained constant, that is, when no new daily tests were reported.

We constructed two databases for our analysis: a primary database, which follows a strict selection rule and serves as the basis for all our main results, and a secondary database that includes all available countries, even if their data were incomplete. This secondary database was used to demonstrate that our findings are not influenced by incomplete data recording within surveillance systems.

The inclusion rule for the primary database was that countries must have sufficient data coverage between May 1, 2020, and January 1, 2022. Specifically, countries were included if they had at least one observation every two weeks (i.e., no more than 14 consecutive days with the same data point), ensuring sufficient variability over the study period. The primary database includes 89 countries, while the secondary database includes 140 countries. Our main results are based on the primary database, while the secondary database was used to confirm the robustness of our findings in the presence of incomplete or delayed data. Countries included in the primary and secondary databases are listed in Supplementary Table 1.

The data used in this study were obtained from OurWorldInData, which provides publicly available and fully anonymized aggregate data on COVID-19-related tests, cases and deaths. In accordance with French regulations governing research involving human data (Loi Jardé, Décret n° 2016-1537), retrospective studies based exclusively on anonymized and publicly accessible datasets do not require approval from a Comit'e de Protection des Personnes (CPP) or an equivalent ethics committee. Therefore, ethical approval was not required for the present analysis.

Moreover, because the study relies solely on publicly available, fully anonymized country-level data, there are no human participants in the usual sense and no individual-level information involved. Such datasets therefore fall outside the scope of human–subjects research under French law, and concepts such as individual informed consent or consent waivers do not apply.

### Other country-level data
Annual population data as of July 1st 2020 and 2021 were sourced from the World Bank national accounts data, along with yearly data on GDP per capita (Current US$) in 2020 and 2021. The Corruption Perception Index (CPI) for 2020 and 2021 was obtained from the Transparency International database. Additionally, the Democracy Index used corresponds to the Democracy Index published by The Economist Intelligence Unit for the years 2020 and 2021. Two health indices were utilized: the WHO health report from 2000 and the CEOWORLD magazine index from 2023, which ranks countries worldwide based on the quality of their health systems.

### Countries of study and regional analysis
Our analysis is primarily conducted at a continental level, with the goal of comparing different regions of the world, particularly focusing on Africa in

relation to the rest of the world. To achieve this, we categorized the countries in our study according to the six regions defined by the World Health Organization (WHO). These WHO regions include the African Region (AFR), the Eastern Mediterranean Region (EMR), the South-East Asia Region (SEAR), the Region of the Americas (AMR), the Western Pacific Region (WPR), and the European Region (EUR). In some cases, we had data from countries that were not listed within the WHO's regional classifications. In these instances, we assigned them to their respective continents based on geographical coherence. For example, in the main database, Liechtenstein was assigned to the European Region (EUR), Guam to the Western Pacific Region (WPR), and Puerto Rico and the United States Virgin Islands to the Region of the Americas (AMR). Similarly, in the secondary database, Anguilla, Aruba, and Curacao were assigned to the Region of the Americas (AMR), while the Northern Mariana Islands were assigned to the Western Pacific Region (WPR).

## Population testing rate: number of tests per 1000 inhabitants

To characterize testing volume, we use the following ratio: the number of tests per 1000 inhabitants over the course of a year (in this case, 2020 and 2021). We refer to this ratio as the "population testing rate". To calculate it, we take the cumulative sum of tests performed in a given year and divide it by the population size in thousands, as reported at the end of that year. For most countries, the number of tests performed in 2020 is taken as the cumulative sum recorded by January 1, 2021. To determine the number of tests performed in 2021, we subtract the cumulative number recorded by January 1, 2021, from the cumulative number by January 1, 2022.

For example, if every inhabitant of a country were tested once in a given year, the population testing rate would be 1000 per 1000 inhabitants for that year. If every inhabitant were tested twice, the rate would be 2000 per 1000 inhabitants. However, as we will demonstrate in our analysis, it is likely that in 2020, the population testing rate will be below 1000 in most countries due to inadequate testing capacity at the onset of the pandemic. We will explore potential correlations between macroeconomic indicators (such as development and governance) and the number of tests per 1000 inhabitants over the course of each year.

We followed a similar method for calculating the positivity rate and case fatality rate. The positivity rate was defined as the ratio of cumulative confirmed cases to the cumulative number of tests performed over each year (2020 and 2021). Specifically, for 2020, the positivity rate was calculated by dividing the cumulative number of cases by January 1, 2021, by the cumulative number of tests by that same date. For 2021, it was calculated by taking the difference in cumulative cases between January 1, 2021, and January 1, 2022, and dividing it by the difference in cumulative tests over the same period.

Similarly, the case fatality rate was determined by dividing the cumulative number of deaths by the cumulative number of confirmed cases for each year. For 2020, this was done by using the cumulative number of deaths and cases as of January 1, 2021. For 2021, we subtracted the cumulative deaths and cases by January 1, 2021, from those recorded by January 1, 2022, and calculated the ratio of these differences.

This approach provides a static snapshot of population testing rates, positivity rates, and case fatality rates in 2020 and 2021, offering valuable insights into the pandemic response across different countries. To further explore the epidemic's dynamics, we analyze acceleration indices for positivity and case fatality rates to assess how these variables evolve over time in response to testing and other factors.

## Acceleration indices for infection and mortality

We study infection and mortality dynamics in WHO regions by measuring the responsiveness of positive tests (cases in the following) with respect to the dynamics of tests and the responsiveness of COVID-19 deaths with respect to the dynamics of positive tests. Measurement is carried out by means of the acceleration index that is defined in ref. [7], in which the main argument put forward is that the number of tests, not time, is the relevant unit of measurement if one is to assess the dynamics of a disease.

When applied to testing and case data, it operates as a test-adjusted reproduction number[9], enabling us to analyze infection dynamics under varying testing conditions.

Essentially, an acceleration index larger (resp. smaller) than unity means that the cumulative positivity rate is increasing (resp. decreasing) over time, indicating that the pandemic is worsening (resp. improving). In Supplementary Method 1, we provide a precise yet broadly applicable mathematical derivation of the acceleration index. We show that it aims at tracking the dynamics and turning points of the cumulative positivity rate for each group of interest (e.g., demographic, spatial), in a way that allows scale-free comparisons between groups and sub-periods. In this paper, we focus the analysis on WHO-defined regions. Crucially, the acceleration index quantifies each region's proximity to pandemic extinction, based on its own testing dynamics: when the acceleration index reaches zero, it signals a temporary halt in new cases—the cumulative positivity rate plateaus—even if testing continues. Cumulative positivity rates have also been used, within given time periods, to evaluate SARS-CoV-2 antibody tests in ref. [11] and to assess vaccine effectiveness in test-negative designs—see, e.g., ref. [12].

Consider first infection analysis and suppose that data is available about the number of tested and positive persons, up to today's date $T$. Denote $\{p_1, \ldots, p_T\}$ the time series of the new (per period) number of positive cases from date $t = 1$ to end date $t = T$. Similarly, $\{d_1, \ldots, d_T\}$ is the time series of new (per period) diagnosed persons. Denote $P_t = \sum_{\tau=1}^{t} p_\tau$ and $D_t = \sum_{\tau=1}^{t} d_\tau$ the numbers of positive and diagnosed persons cumulated up to date $t$. Note that by definition, $P_t$ and $D_t$ are positive numbers that are non-decreasing over time since $p_t$ and $d_t$ are non-negative numbers. The acceleration index, denoted $\varepsilon_T$ at date $T$, is an elasticity that measures the proportional responsiveness of cumulated cases with respect to cumulated tests. Given that the number of cases and tests are not necessarily varying at the same growth rate across time within WHO regions, the acceleration index measures the percentage change of cases within a region with respect to its own percentage change of tests and is thus unit-free. The acceleration index is defined as follows with weekly data:

$$\varepsilon_T = \left[\frac{P_T - P_{T-1}}{P_T}\right] \div \left[\frac{D_T - D_{T-1}}{D_T}\right] = \left[\frac{p_T}{P_T}\right] \div \left[\frac{d_T}{D_T}\right] \quad (1)$$

Rearranging the terms of the latter equality, we see that the acceleration index relates to the weekly and average positivity rates, in the following way:

$$\underbrace{\varepsilon_T}_{\text{acceleration index}} = \underbrace{\frac{p_T}{d_T}}_{\text{weekly positivity rate}} \div \underbrace{\frac{P_T}{D_T}}_{\text{average positivity rate}} \quad (2)$$

While $D_t$ and $P_t$ become strictly positive the first period some testing is performed and some persons are tested positive, it could happen that $d_t = 0$ for some later periods, in which case the positivity rate $p_t/d_t$ is not defined for those periods. In our weekly dataset, this configuration did not occur, though. Compared to Supplementary Method 1,

we here term the cumulative positivity rate $P_T/D_T$ the average positivity rate because it admits the following decomposition:

$$\frac{P_T}{D_T} = \sum_{\tau=1}^{T} \frac{p_\tau}{d_\tau} \frac{d_\tau}{D_T} \quad (3)$$

Equation (3) demonstrates that the cumulative positivity rate can be defined as a weighted average of weekly positivity rates, with each week's weight equal to its share of total tests since the pandemic began. This approach rightly assigns greater influence to weekly positivity rates with higher testing volumes, as they are more informative.

Equation (2) shows that the acceleration index is the elasticity of cumulative cases with respect to cumulative tests, potentially measured over long epidemic periods. Importantly, for real-time surveillance, the index must be recalculated each time new testing and case data become available.

When tests are time-varying—as observed for most countries during COVID-19—the dynamics of cases must be adjusted for the dynamics of tests to accurately measure whether the pandemic is intensifying or abating. This is exactly what our acceleration index does. Although in any period the number of weekly cases $p_t$ must be by construction smaller than, or equal to, the number of weekly tests $d_t$, so that the weekly positivity rate $p_t/d_t$ is smaller than or equal to one, the acceleration index is not bounded above by one. In fact, it signals either acceleration, when larger than one, or deceleration, when smaller than one. In the former case, cumulated cases grow faster than cumulated tests, while in the latter case, cumulated tests grow faster than cumulated cases.

When $\varepsilon_T > 1$, hence, the average positivity rate $P_t/D_t$ increases, i.e., the pandemic is intensifying: in that case, increasing cumulated tests by 1% leads to *more than* 1% of cumulated cases. Accordingly, a valid public health objective is to drive the acceleration index below one: this would indicate that the pandemic decelerates and becomes under control. Ideally, one would like to find ever fewer cases the more one tests, proportionally, and to see $\varepsilon_T$ approach zero: the acceleration index equals zero when the growth rate of cumulated cases also reaches zero, indicating that the pandemic has stopped producing new cases, even if temporarily so.

Our approach further demonstrates that monitoring current positivity rates $p_t/d_t$ alone is inadequate: they capture infection speed but not its acceleration and lack unit-free comparability, and thus are ill-adapted for cross-region analysis. Moreover, fluctuations in the instantaneous positivity rate $p_t/d_t$ capture short-term trends (e.g., weekly) that may diverge from the trajectory of the cumulative positivity rate $P_t/D_t$. For example, the instantaneous positivity rate may fall this week relative to last, yet an acceleration index above one would still signal that the epidemic is intensifying. Formally, this would occur when $p_{t-1}/d_{t-1} > P_t/D_t$ and $p_t/d_t$ lies in between the two values involved in the latter inequality. One would visualize such a configuration in the data when the weekly positivity rate goes down while still being larger than the average positivity rate: in that case, our approach would say that although this week's positivity rate has declined compared to last week, the pandemic is still in an acceleration regime, as this week's positivity rate remains larger than the average positivity rate since the beginning of the pandemic, indicating at best *a decreased acceleration*. Similarly, a weekly positivity rate that increases could be consistent with a deceleration regime signalled by an acceleration index that, while increasing, still stays smaller than one.

Lastly, in the hypothetical case where the weekly positivity rate $p_t/d_t$ remains constant over time, the acceleration index converges to one, marking the threshold between epidemic acceleration and deceleration. As illustrated in ref. 7, the index can also be intuitively displayed on a scatter plot of cumulative cases versus cumulative tests, each normalized to its latest value.

The acceleration index defined above to capture the responsiveness of cases with respect to tests is easily adapted to measure the dynamics of mortality. Reasoning along similar lines, one concludes that the dynamics of cases should be taken into account so as to properly measure the dynamics of fatalities. Because the number of positive cases is typically time-varying, the acceleration index offers a theoretically sound way to measure the mortality dynamics adjusted for the case dynamics. By substituting variables, we can define an analogous index that measures the elasticity of cumulated deaths with respect to cumulated cases. Denote $\{m_1, \ldots, m_T\}$ as the time series of new deaths (or mortal health outcomes) per period, and let $M_t = \sum_{\tau=1}^{t} m_\tau$ represent the cumulative number of deaths up to date $t$. The severity acceleration index at date $T$ is given by:

$$\eta_T = \left[\frac{M_T - M_{T-1}}{M_T}\right] \div \left[\frac{P_T - P_{T-1}}{P_T}\right] = \left[\frac{m_T}{M_T}\right] \div \left[\frac{p_T}{P_T}\right] \quad (4)$$

provided that $p_T$ and $M_T$ are strictly positive. The acceleration index $\eta_T$ then measures the percentage change in cumulated deaths relative to the percentage change in cumulated cases at date $T$, offering insights into the acceleration of severity as the epidemic evolves once the dynamics of

infection have been corrected for. Similar to the acceleration index for cases and tests, a value of $\eta_T$ greater than one would indicate that the severity of the epidemic, in terms of deaths relative to cases, is accelerating, whereas a value smaller than one would suggest a deceleration in mortality severity. Here again, if the case fatality rate $m_t/p_t$ were constant over time, $\eta_t$ would be equal to one at all dates, unlike what is seen in the data for most regions. In addition, it may help intuition to think of $\eta_T$ as the ratio of the current (here, weekly) case fatality rate to its average since the starting period for the data. The acceleration index of cumulated deaths to cumulated tests, similarly defined as the product of $\varepsilon_T$ and $\eta_T$, therefore mixes, by contrast, the acceleration of both infection and severity.

## Results
### Population testing, positivity and case fatality rates in 2020 and 2021

Figure 1 presents yearly numbers of tests performed by countries grouped by regions for 1000 inhabitants, i.e., the population testing rate (see Supplementary Table 2 for descriptive statistics). Panel (a) depicts results for the year 2020. It shows that testing population rates vary sharply between regions (Kruskal–Wallis $\chi^2$, $p < 0.0001$) with a factor of about 16. Median testing rates for 1000 inhabitants range from 26 tests to 431, hence a factor of 16.6 between regions. The African Region (AFR) presents the lowest testing range, statistically different from all other regions (Wilcoxon rank sum test, $p < 0.0001$), whereas the median testing rate is the highest for countries in the European Region (EUR) (Wilcoxon rank sum test, $p < 0.0001$). Countries in the Eastern Mediterranean Region (EMR), Western Pacific Region (WPR) and the Region of America (AMR) are at an intermediate level with 204, 126, and 107 tests per 1000 inhabitants, respectively. South-East Asian Region (SEAR) remains at a low level, close to that observed in Africa, with 59 tests per 1000 inhabitants in 2020.

In 2021 (panel b), population testing rates have increased for all regions, but at different paces. African countries exhibit a median yearly testing rate 288% higher in 2021, but this remains at a low level, with only 101 tests per 1000 inhabitants. It remains the lowest median testing rate observed in 2021. Median testing rates have increased by 151% in the Western Mediterranean Region (514 tests per 1000 inhabitants) and 308% in the South East Asian Region (241 tests per 1000 inhabitants). Median testing rate in the Region of the Americas has increased by 320% to establish at 450 tests per 1000 inhabitants and by 235% in North America and Europe. In 2021, it remains the highest median testing rate with 1448 tests per 1000 inhabitants. The Western Pacific Region has seen the highest increase in median testing rate (519%), making it one of the highest regions in terms of testing capacity, with 781 tests per 1000 inhabitants in 2021. All in all, although median testing rates have sharply increased for all regions, disparities remain very high in 2021, with a factor of about 14.3 between African countries, still the lowest rate observed (Wilcoxon rank sum test, $p < 0.0001$), and the European Region, the highest rate observed (Wilcoxon rank sum test, $p < 0.0001$). This highlights the necessity to take tests into account in order to better understand the transmission and mortality between regions.

Figure 2 presents the distributions of the number of cases as a fraction of tests by year and countries grouped in regions, i.e., positivity rates in % (see Supplementary Table 3 for descriptive statistics). In 2020, the Region of the Americas saw the highest mean positivity rate: the median regional rate was 18.9%. The positivity rate in American countries is far higher than in other countries, making the median positivity rates of American countries significantly different from the other regions ($p = 0.0003$, Wilcoxon Rank Sum Test). They were followed by European countries, Eastern Mediterranean countries, African countries and South-East Asian countries that exhibit similar patterns for their test results: median is 8.6%, 8.5%, 6.5%, and 5.3% respectively ($p = 0.3283$, Krustal-Wallis test). Western Pacific countries follow with a median value of 2.5%, significantly different from the other regions ($p = 0.0089$, Wilcoxon Rank Sum Test), which appears to be the region with the lowest number of cases as a fraction of tests.

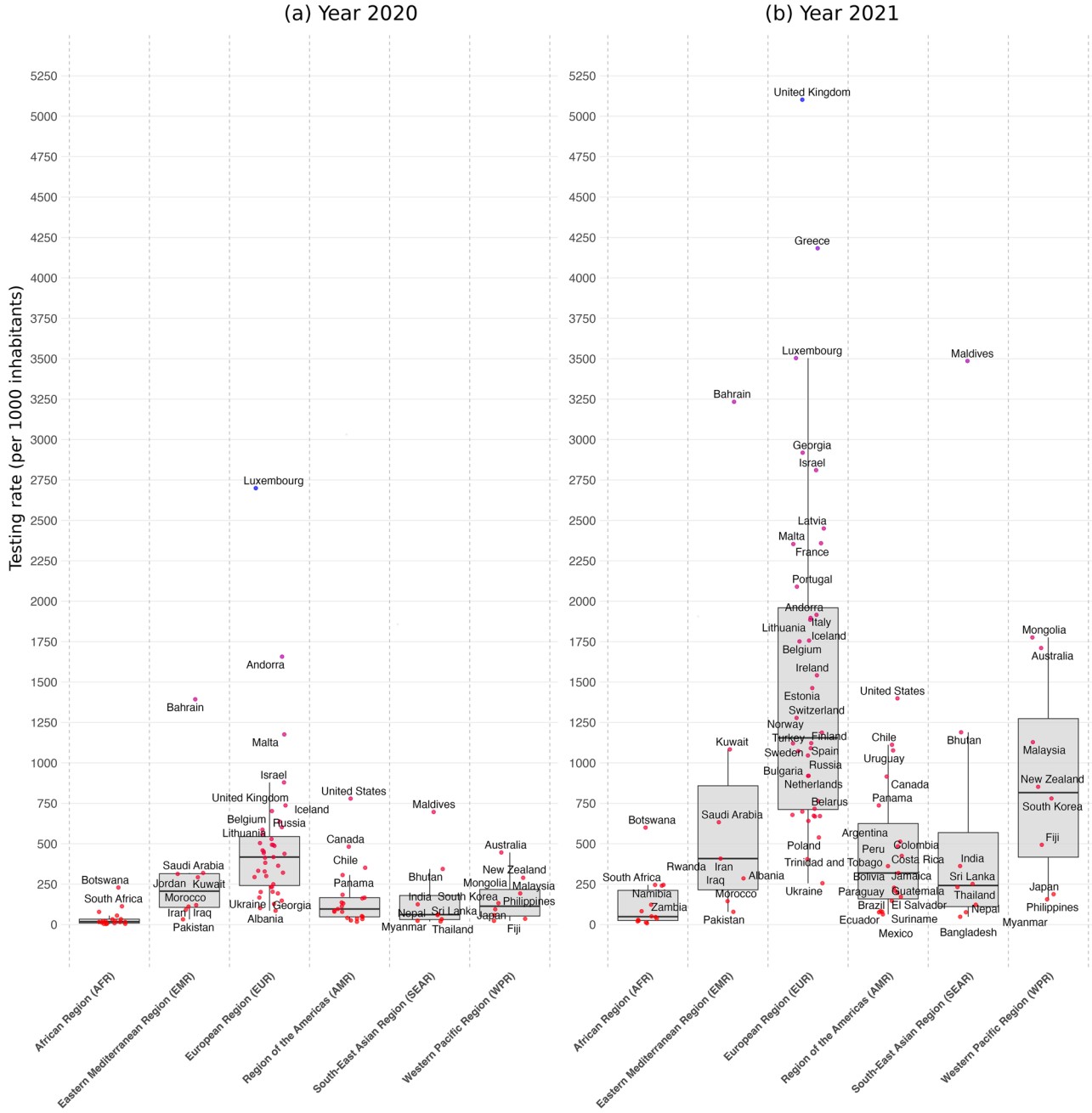

**Fig. 1 | Yearly numbers of COVID-19 tests performed per 1000 inhabitants by countries grouped in regions, for 2020 and 2021.** Panel **a** presents Box plots for yearly numbers of COVID-19 tests by countries grouped by regions, for the year 2020 ($n = 89$). The box is drawn from first to third quartile with a horizontal line drawn inside it to denote the median. Whiskers based on the 1.5 IQR value. Panel **b** presents Box plots for yearly numbers of tests by countries grouped by regions, for 2021 ($n = 89$). For display purposes, five outlier countries were removed in 2021. Austria, with a testing rate of 13,562 in 2021, Cyprus, with a testing rate of 15,131 in 2021, the United Arab Emirates, with a testing rate of 9119 in 2021, Slovakia, with a testing rate of 7972 in 2021 and Denmark, with a testing rate of 7333 in 2021.

Although American countries remain the countries with the highest proportion of positive (median value is 14.1%), the pattern observed in 2020 for other regions does, however, not persist. Eastern Mediterranean countries have seen a large decrease in cases as fraction of tests: the median value in 2021 was 5.4 as compared to 8.5% in 2020 (paired wilcoxon tests, $p = 0.0078$), while South-East Asian countries have seen a large increase in cases as fraction of tests: the median value in 2021 was 11.5% as compared to 5.3% in 2020 (paired wilcoxon tests, $p = 0.4375$). Similarly, the Western Pacific Region saw a significant increase in positivity rate: the median value in 2020 was 2.5% as compared to 6.1% in 2021 (paired Wilcoxon tests, $p = 0.1563$), and they are on par with other regions such as Africa (8.8%), Eastern Mediterranean Region (5.4%), and European Region (6.1%).

All regions faced a significant number of cases as a fraction of tests, and this is particularly true for the year 2021.

Figure 3 presents the yearly distribution of deaths as a fraction of cases by countries grouped by regions, i.e., case fatality rates in % (see Supplementary Table 4 for descriptive statistics). Countries in the African Region, Eastern Mediterranean Region, South-East Asian Region, Western Pacific Region, and European Region exhibited a median yearly case fatality rate between 1% and 3%. Median yearly mortality rate in 2020 was 2.6% in the Region of the Americas, significantly different from other regions ($p = 0.0003$, Wilcoxon Rank Sum Test). The results exhibit significant differences in yearly case fatality rates between regions ($p = 0.0092$, Kruskal–Wallis test).

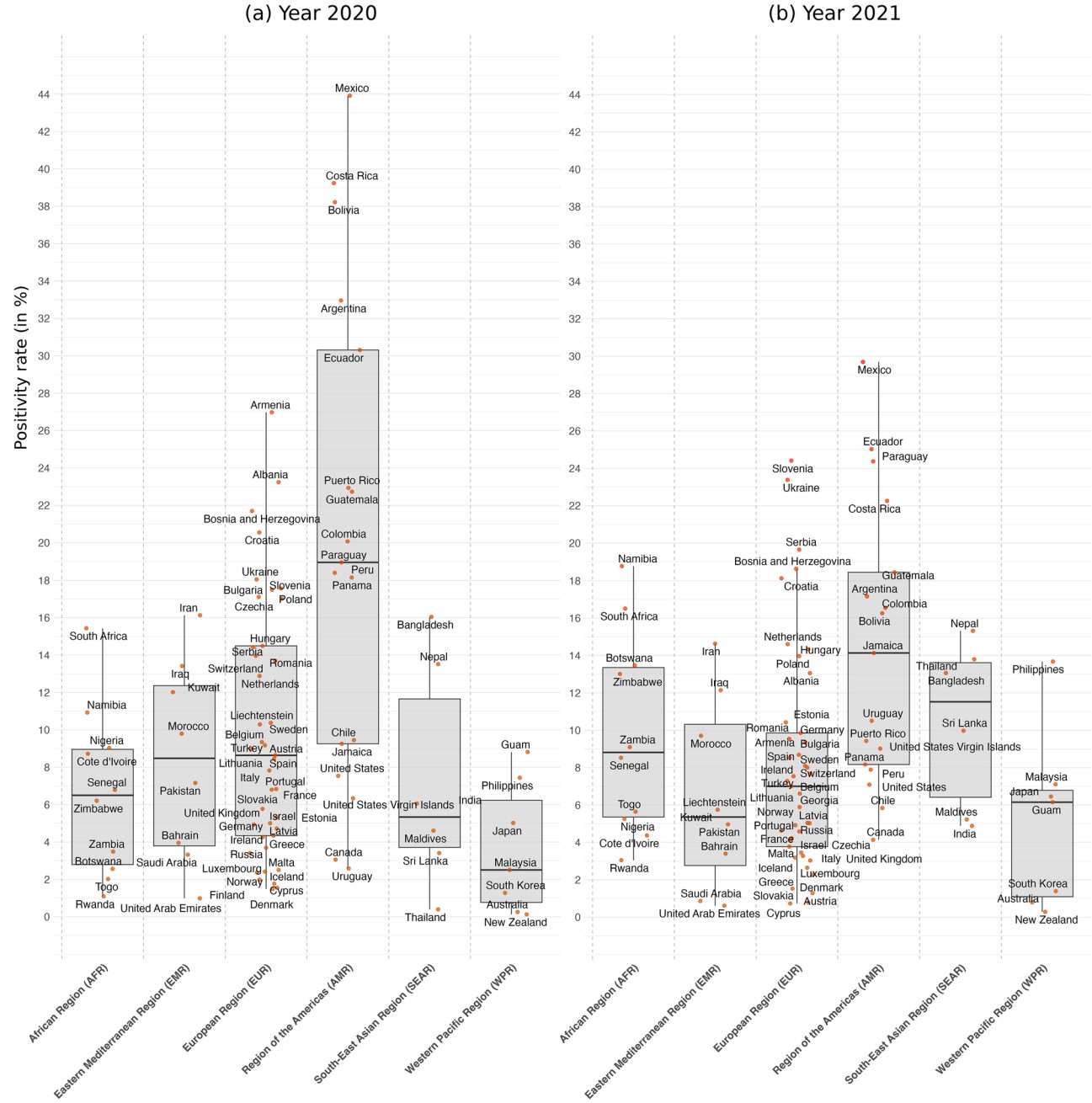

**Fig. 2 | Yearly COVID-19 positive cases as a fraction of tests—i.e., positivity rates in %—by countries grouped in regions, for 2020 and 2021.** Panel **a** presents Box plots of the number of COVID-19 positive cases as a fraction of tests by countries grouped in regions for 2020 (*n* = 89). The box is drawn from first to third quartiles with a horizontal line drawn inside it to denote the median. Whiskers based on the 1.5 IQR value. Panel **b** presents Box plots of the number of positive cases as a fraction of tests by countries grouped in regions for 2021 (*n* = 89).

In 2021, panel (b) shows that countries in Africa, South-East Asia, the Middle East, and the Americas have seen a steady yearly case fatality rate (*p* = 0.02667, *p* = 0.1953, *p* = 0.4375, and *p* = 0.4316, respectively, paired Wilcoxon rank sum test). Countries in Europe have seen a 44 percentage points decrease in COVID-19 mortality with a median mortality rate under 1% (*p* = 0.01271, paired Wilcoxon rank sum test). South-East Asia remains an exception in 2021 with an increasing median yearly fatality rate, while other regions' median yearly fatality rate is decreasing from one year to the next.

**Acceleration of infection and of mortality across WHO regions**

Although the annual data depicted in Figs. 1–3 is relatively coarse, it shows strong disparities between regions and reveals that Africa experienced

significantly lower testing rates, larger test-positivity and case-fatality ratios over 2020 and 2021. These preliminary findings substantiate our central hypothesis and constitute the groundwork for the more detailed, weekly-level, dynamic analysis of infection and case fatality rates that follows. We present in Fig. 4 the acceleration indices for infection (in blue) for WHO regions, including raw data acceleration indices, non-parametric estimates, and their respective confidence intervals. These estimations provide two key insights. First, the confidence intervals depicted in Fig. 4 indicate whether we can reliably interpret the raw acceleration indices. Second, and most importantly for our analysis, the estimated confidence intervals enable the identification of statistically significant acceleration episodes in infection and mortality (i.e., acceleration indices exceeding 1), which are of critical importance in this study. These confidence intervals are derived from data

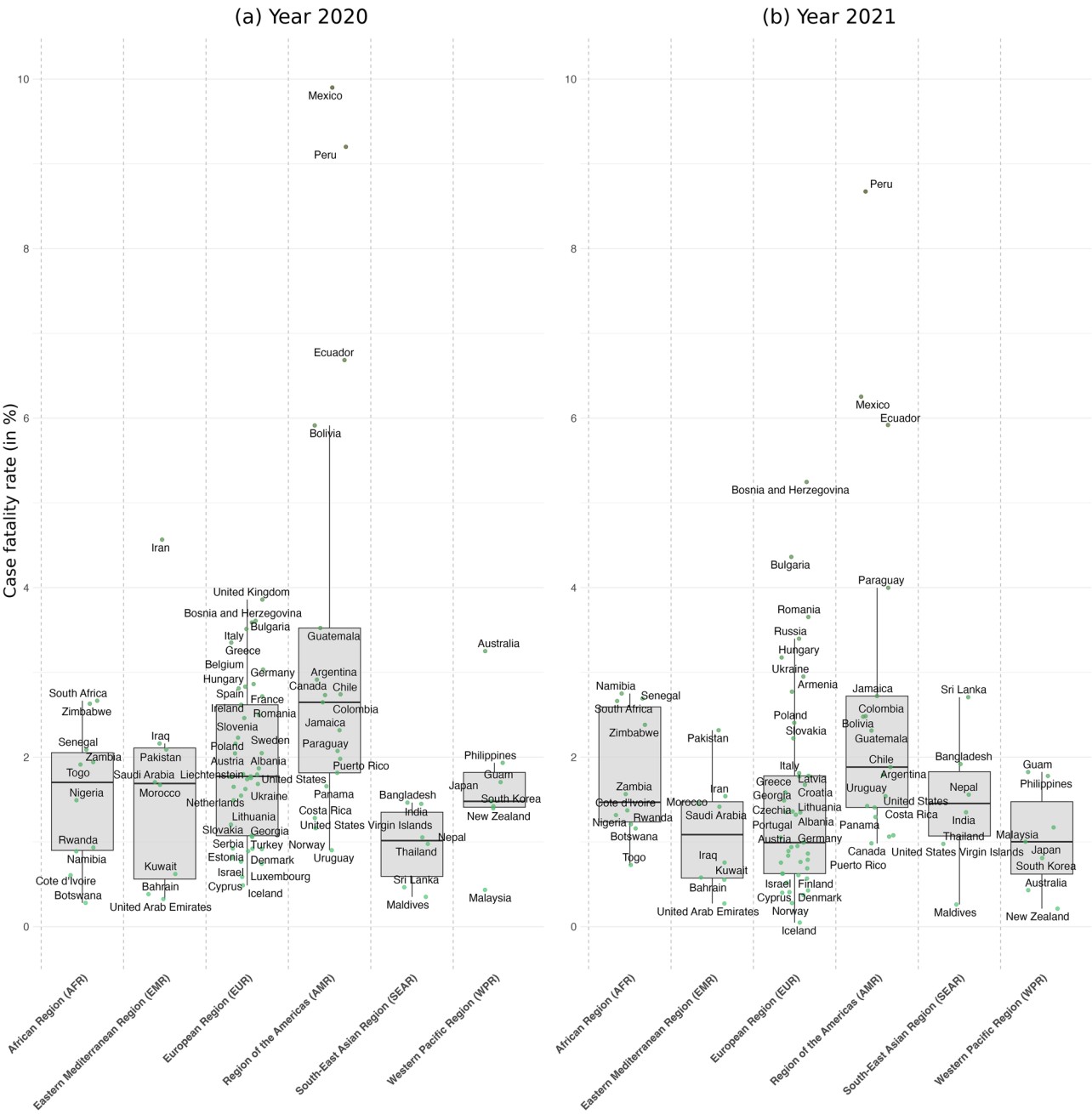

**Fig. 3 | Yearly COVID-19 deaths as a proportion of cases, i.e., case fatality rates in %, by countries grouped by regions, for 2020 and 2021 ($n = 89$).** Panel **a** presents Box plots of yearly COVID-19 deaths as a fraction of case rates by countries grouped in regions for 2020 ($n = 89$). The box is drawn from first to third quartile with a horizontal line drawn inside it to denote the median. Whiskers based on the 1.5 IQR value. Panel **b** presents Box plots of yearly mortality rates by countries grouped in regions for the year 2021 ($n = 89$).

aggregated at the regional (WHO) level and thus reflect not only statistical uncertainty but also substantial heterogeneity across the countries included in each region. This heterogeneity stems from differences in epidemic trajectories, testing practices, reporting quality, and health system responses. The width of the confidence intervals should therefore not be interpreted solely as a sign of imprecision, but rather as an indicator of this underlying variation. By reporting these intervals, we aim to transparently account for this heterogeneity and still demonstrate that the African Region, despite such internal differences, experienced statistically significant acceleration episodes in both infection and mortality.

Curves in blue and their respective confidence intervals indicate that all regions have known episodes of acceleration of infection, with cases growing faster than tests (indicated by an acceleration index greater than one). This is

particularly true for the African region, which exhibits up to four distinct acceleration episodes: (1) the second and third quarters of 2020, (2) the fourth 2020 quarter and the first 2021 quarter, (3) the second and third quarters of 2021, and (4) end of 2021 fourth quarter. Maximum estimated acceleration observed at the peak of the episodes ranges from 1.92 (95% CI [1.42–2.61]) to 2.17 (95% CI [1.60–2.97]), indicating that when tests increased by 1%, cases increased by around 2% at worst. The Western Pacific Region has also experienced several significant episodes of acceleration: we can also count four distinct acceleration episodes during 2020 and 2021. Maximum acceleration in these episodes ranges from 1.80 (95% CI [1.32–2.49]) to 2.10 (95% CI [1.56–2.85]). The Europe Region has experienced a first episode at the beginning of 2020 with a maximum estimated acceleration at 1.61 (95% CI [1.32–1.74]). It lasted around one month.

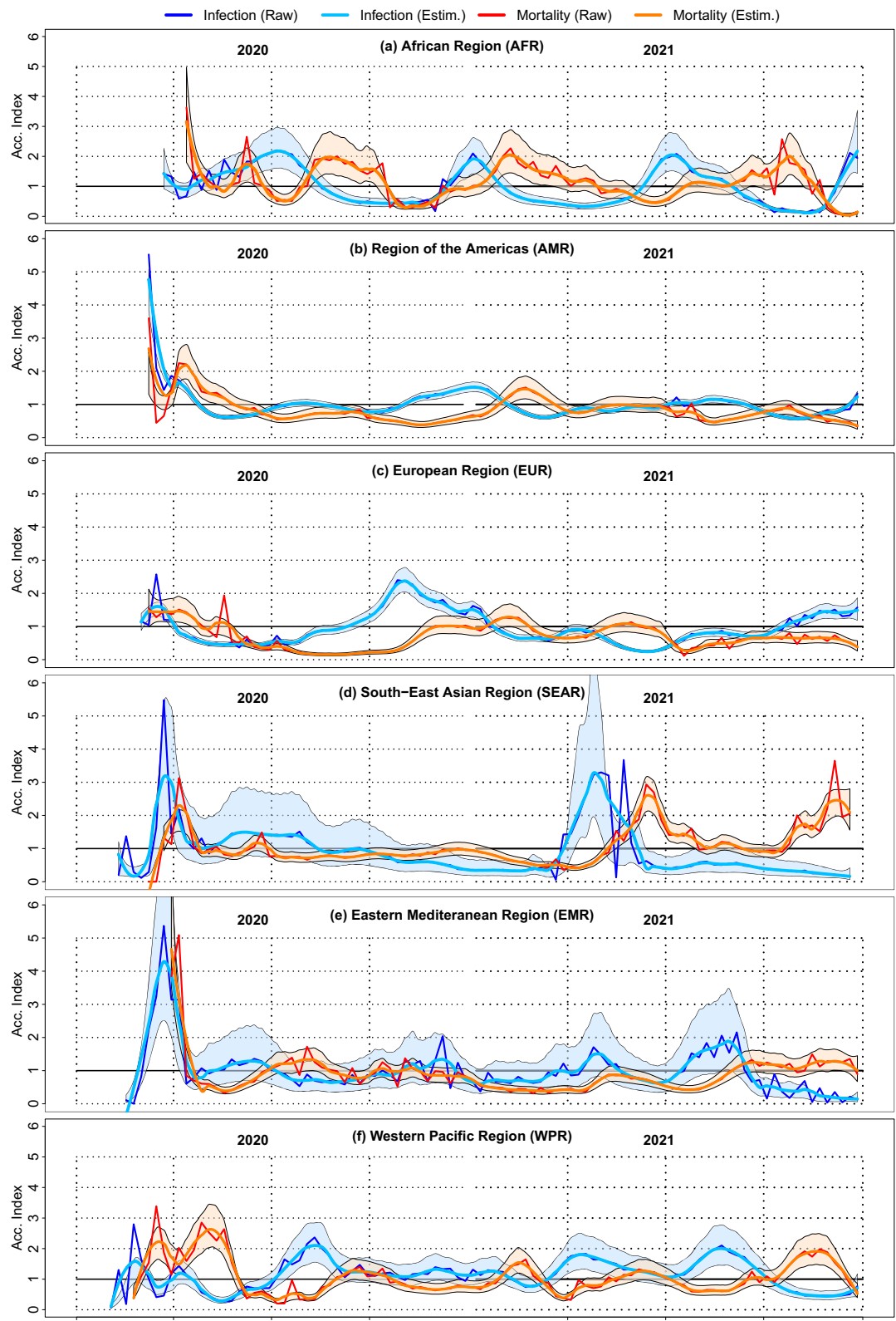

**Fig. 4 | COVID-19 acceleration indices for infection and mortality in WHO regions, for 2020 and 2021.** Each graph depicts raw and estimated acceleration indices for COVID-19 infection and mortality by WHO regions. Estimations are carried out using locally weighted polynomial regressions (LOESS). Acceleration index for infection based on raw data is depicted with a deep blue (resp. red for the acceleration index for mortality) and its LOESS estimate in light blue (resp. orange for the estimated acceleration index for mortality)—see legend. Blue (resp. red) shaded areas indicate 95% confidence intervals of infection (resp. mortality) esti-mated indices. Smoothed confidence bands are estimated on the log-scale—using the exact same dates—and back-transformed into the original metric. Lower bounds are strictly positive by construction. Each year is divided into quarters to facilitate reading.

During fall 2021, we observed a more stringent episode with estimated maximum acceleration at 2.37 (95% CI [2.04–2.78]) and a third wave during the fourth 2021 quarter, but with an estimated acceleration of cases at 1.49 (95% CI [1.18–1.87]), still accelerating at the end of the period of interest. Estimation for the South-East Asian region is less accurate as shown by a larger confidence interval, but still suggests two statistically significant acceleration episodes with maximum estimated acceleration at 3.19 (95% CI [1.52–5.52]) in the second 2020 quarter and 3.29 (95% CI [1.97–7.21]) in the second 2021 quarter. This means that during these two episodes, cases were growing up to three times faster than tests. The American region has experienced a large acceleration episode at the beginning of the observation period with a maximum estimated acceleration at 4.77 (95% CI [3.81,5.08]) and a second, but less important peak with a maximum acceleration index at 1.52 (95% CI [1.38–1.68]). Finally, only one acceleration episode appears as significant during the third 2021 quarter for the Eastern Mediterranean Region–the maximum estimated acceleration index is 1.89 (95% CI [1.11–1.93]).

Overall, measures of community-wide synchrony indicate that the observed dynamics of tests and cases undergo low but significant correlated waves when all regions are considered. Community synchrony is 0.3859 (normalized between 0 and 1 is 0.3216) with $p = 0.0004$. Computing community-wide synchrony for each year shows that overall correlation between infection dynamics was higher in 2020 (community synchrony = 0.4274, normalized 0.3562, $p = 0.0079$) than in 2021 (community synchrony=0.2854, normalized 0.2379, $p = 0.0068$).

Pairwise dependence measured by Kendall correlation coefficients over the whole period shows that among the 15 pairs, one third of them (5) exhibit a statistically significant correlation coefficient at a 5% $p$-value (see details in Supplementary Table 5). Infection acceleration episodes are therefore interrelated over the WHO regions for the considered period, but as indicated by community synchrony analysis and Kendall correlation, transmission dynamics are not fully synchronized. In other words, there is not a single transmission dynamics shared by all WHO regions; rather, different waves of infection acceleration and deceleration.

In summary, two conclusions follow from Fig. 4. First, Africa was not spared from COVID-19 when the dynamics of testing are factored into the analysis, allowing for a more accurate account of positive cases and deaths. Second, Africa experienced, in reality, a *larger* number of episodes during which infection accelerated at a similar magnitude, relative to all the other regions. In particular, when viral acceleration peaked, increasing the number of cumulated tests by 1% meant an increase in the number of cumulated positive cases of about 2%, which turns out to be similar in magnitude to infection acceleration peaks in other regions. In addition, what is striking is that in Africa, the magnitude of the peaks in mortality acceleration is also about 2, meaning that increasing the number of cumulated cases by 1% meant an increase in the number of cumulated deaths of about 2% at peak. Similar sequences of infection-acceleration followed by mortality-acceleration with comparable magnitudes are also observed in the WPR and SEAR; however, in the African Region, these episodes occur with even greater frequency.

### Coupling between infection and mortality accelerations

Figure 4 also provides the estimated acceleration indices for case fatality rates across WHO regions with their respective 95% CI (curves and shaded areas in red). CI shows that estimations provide good estimates of case fatality dynamics for all regions except the Eastern Mediterranean Region, for which, after the first acceleration episode, CI remains large, and none of the fluctuations are significantly above 1. All other regions have experienced at least one statistically significant acceleration case fatality acceleration episode, indicating that COVID-19 deaths were growing faster than cases.

Graphical examination suggests a coupling between transmission and mortality: acceleration episodes for infection are often followed by a delayed acceleration in mortality. This also seems true for deceleration episodes. This phenomenon is particularly salient for the African Region, but we can see it, although to a lesser extent, for other regions. In Fig. 5, we examine

more systematically the coupling between delayed transmission and mortality by plotting lags of the transmission acceleration index against the current mortality acceleration index. Each plot displays the relationship between the lagged infection acceleration index (X-axis) and the current case fatality acceleration index ($y$-axis) by WHO region. This visualization allows us to assess how past changes in infection correlate with current changes in case fatality rates. Each line of plots corresponds to a specific lagged infection acceleration index, ranging from 1 week (top row) to 16 weeks (bottom row). In each plot, we estimate a non-parametric spline (in red) to illustrate the relationship between these two variables. When the line is flat, the correlation is zero. Additionally, we provide a quantitative measure of correlation using the Kendall correlation coefficient.

Graphical examination and Kendall analysis indicate that this infection/mortality coupling is present for four regions out of six: AFR, AMR, EUR and WPR. Maximum statistically significant correlation between delayed transmission and mortality acceleration indices is observed for lags between 7 to 9 weeks in all four regions—the Africa region and the Europe Region exhibiting the largest Kendall correlation. Between weeks 7 and 9 of delay, both deceleration and acceleration episodes in transmission and mortality are aligned as shown by the non-parametric regression line (in red).

What differs, however, is that the coupling between transmission and mortality indices fades out between week 12 and 14 for all regions but Europe. This is because the correlation between transmission and mortality remains for acceleration indices below 1 (seen in the bottom left quadrant of each plot), whereas this correlation disappears for other regions.

Figure 5 reveals that Africa is the region that has seen the most severe form of coupling between infection and mortality. When lagged infection acceleration is observed, it can translate almost one for one into mortality acceleration in the next seven to ten weeks: a 1% increase in tests leads to a 2% increase in cases during acceleration episodes and to a delayed 2% increase in case fatality rate. The combined effect indicates that when tests increased by 1%, COVID-19 deaths increased by 4%, 7–10 weeks later in the Africa Region during acceleration episodes. Moreover, it takes a lagged infection acceleration significantly larger than one (on the $x$-axis) for mortality to accelerate in SEAR and WPR, whereas for AFR, mortality accelerates as soon as infection is approaching the acceleration regime. In sharp contrast, mortality either does not accelerate or even *decelerates* whenever infection accelerates in AMR, EUR and EMR. This strongly suggests that Africa handled the surges in infection in terms of mortality.

## Discussion

As of December 31, 2021, Africa reported 4,708,561 confirmed COVID-19 cases and 112,608 deaths, figures significantly lower than in Europe, which recorded 94,734,053 cases and 1,635,574 deaths—roughly 20 times more cases and 15 times more deaths in Africa. However, testing is the primary tool for identifying cases, and without adequate testing, both cases and deaths may be underreported, leading to an incomplete picture of the true spread of the virus[13–16].

Our analysis shows that population testing rates in Africa—26 tests per 1000 inhabitants in 2020, rising to 101 per 1000 in 2021—were insufficient to capture the true number of cases and deaths. In 2020, positivity rates were comparable between Africa and Europe. The African Region (AFR) showed a median positivity rate of 6.5%, close to the European Region (EUR) rate of 8.6%. In some African countries, higher positivity rates suggest substantial under-reporting of cases and more widespread transmission than official numbers reflect. The case fatality rate (CFR), which measures the proportion of deaths among confirmed cases, varied across regions, likely reflecting differences in healthcare capacity and reporting[17]. The median CFR in Africa in 2020 was 1.7%, slightly lower than Europe's 1.8% and similar to the global average.

By 2021, disparities in testing between regions became more pronounced. Europe increased its testing significantly, reaching a median of 1448 tests per 1000 people, compared to just 101 tests per 1000 in Africa. This difference resulted in continued underreporting of cases in Africa,

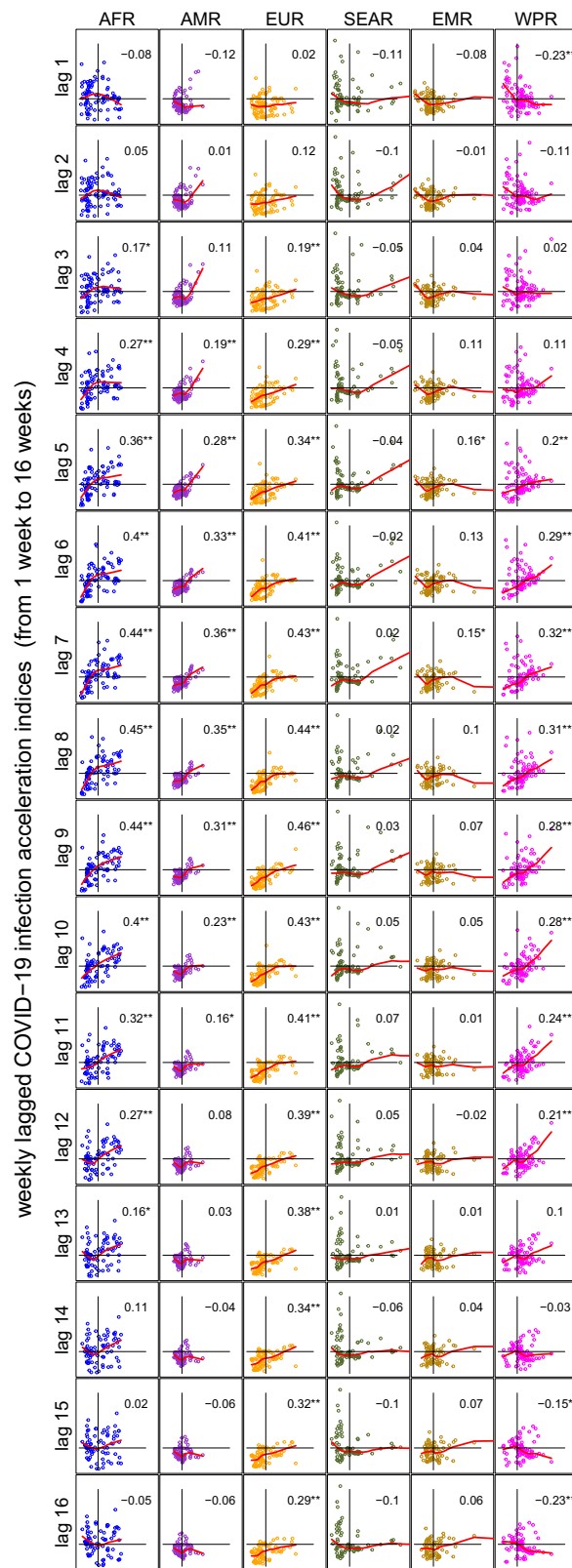

**Fig. 5 | Coupling between weekly COVID-19 infection acceleration index (lagged) and mortality acceleration index across WHO regions.** Each panel presents a scatter plot of weekly lagged COVID-19 infection acceleration indices on the *x*-axis and the weekly current case fatality acceleration index on the *y*-axis. *xy*-axes range between 0 and 4 (labels are omitted). The horizontal and vertical black lines in each plot delimit the value 1 so that the upper right quadrant corresponds to lagged infection and mortality accelerating, and the bottom left to lagged infection and mortality decelerating. In each panel, we estimate a locally-weighted polynomial regression (in red) and provide the Kendall correlation coefficient and its associated *p*-value. Each row corresponds to lagging (from one to 16 weeks) the infection acceleration index, and each column to a WHO region.

cases. In contrast, Africa's relatively stable CFR may reflect limited testing and healthcare resources, where only the most severe cases were detected, leading to higher fatality rates.

Our dynamic analysis using acceleration indices supports these findings by relating testing, cases, and deaths over time. This method reveals that Africa experienced multiple episodes of infection acceleration, where increases in testing were followed by sharp rises in reported cases, underscoring periods of intense virus spread. The emergence of new variants—such as Beta, Delta, and Omicron—corresponded to these waves of epidemic acceleration. During these peaks, the acceleration index reached a value of 2, meaning that for each 1% increase in testing, there was a 2% increase in reported cases. These peaks in infection coincided with major events, such as the first epidemic wave in mid-2020, the spread of the Beta variant in late 2020, and the rise of the Delta and Omicron variants in 2021[18–20].

In comparison, other WHO regions experienced less intense infection acceleration. This was particularly evident in the European Region and the Region of the Americas. The South-East Asia Region initially delayed viral transmission in 2020, likely due to stricter public health measures[21], but experienced the highest acceleration spikes of all WHO regions in 2021.

Moreover, we document a significant coupling between infection and mortality in Africa, with a notable time lag of about 8 weeks between peaks in case acceleration and subsequent peaks in death acceleration. This lag indicates that as cases surged, mortality rates increased a few weeks later, likely due to healthcare systems being overwhelmed by the surge in severe cases. Similar dynamics were observed in Lombardy, Italy, where ICU capacity was exceeded during the first wave of COVID-19, leading to delayed care and higher mortality rates[22]. In Africa, where healthcare infrastructure is more limited, this delay was even more pronounced.

The observed 8-week lag underscores the importance of timely public health interventions. The relationship between case surges and mortality suggests that improving healthcare capacity and accelerating medical responses during periods of high transmission could significantly reduce fatalities. Strengthening testing, treatment, and healthcare systems is critical to managing future viral waves and minimizing the pandemic's death toll. While these insights are a particularly urgent matter for the African Region —the primary focus of our study—we also observe analogous infection-mortality acceleration sequences in the WPR and SEAR, underscoring that timely testing and health-system strengthening are broadly critical across WHO regions.

In this study, we adopt an epidemiological approach by analyzing the dynamic relationship between testing and case counts, allowing us to identify periods of infection acceleration and their association with mortality acceleration across WHO regions. These regions include countries with varying levels of GDP, which may have influenced their testing capacities. Using econometric models that account for heteroskedasticity and regional differences, we examined how testing rates were influenced by economic wealth (GDP), healthcare quality (measured by WHO or CEO health indices), and institutional quality (measured by corruption or democracy indices). Our findings show that in 2020, GDP was the only statistically significant factor affecting testing rates. However, by 2021, none of these indices remained statistically significant (see Supplementary Note 1 for further details).

despite ongoing viral transmission. In 2021, Africa's median positivity rate was 8.8%, higher than Europe's 7%. The median CFR in Africa slightly decreased to 1.5%, while Europe saw a more marked decline to 1%. Other regions, such as the Americas, also experienced notable reductions in CFR, which can be attributed to improved medical interventions, higher vaccination rates, and more comprehensive testing efforts that identified milder

Our study has several other limitations. The first limitation is the delay in processing and reporting test results, particularly in overburdened health systems. These delays can result in outdated data that does not accurately capture the real-time spread of the virus. In many regions, overwhelmed reporting systems contributed to data gaps, making accurate assessments difficult. We addressed this issue by using weekly data rather than daily data, as daily data are more susceptible to processing delays. Additionally, we selected only monotonic changes in cumulative tests, cases, and deaths from the original dataset to ensure consistency. A second limitation of our study is that data on tests, confirmed cases, and COVID-19-related deaths are only available for a subset of countries in the African Region. However, these countries are not statistically different from other African nations in terms of GDP per capita or population size (mean difference test, $p = 0.6704$ and $p = 0.4411$, respectively). In addition, a more comprehensive analysis—incorporating all available data, including those with reporting delays (see secondary database in the "Methods" section)—more than doubles the number of included countries by relaxing the strict selection criteria. This expanded analysis yields the same conclusions (detailed in Supplementary Figs. 3 and 4), further reinforcing the robustness of our findings.

A third limitation is the underestimation of asymptomatic infection. Since many asymptomatic individuals were never tested, relying solely on testing data likely underestimates the true scope of transmission. This may also affect the accurate assessment of mortality dynamics. Another limitation is test accuracy, particularly the issue of false negatives. RT-PCR tests can miss cases due to factors such as improper sample collection, low viral loads, or testing at the wrong time in the infection cycle (either too early or too late[23]). This underdiagnosis further skews case data. However, both of these limitations point towards an underestimation of infection, and our findings already demonstrate that the Africa WHO region was no exception in this regard.

Lastly, many countries initially prioritized testing symptomatic individuals or high-risk groups, which neglected asymptomatic carriers who could still spread the virus. This approach likely led to an underestimation of the true number of cases, which again strengthens our results. As the pandemic progressed, testing policies shifted (e.g., expanding from symptomatic testing to broader surveillance testing), which complicates comparisons of case data across different time periods or regions. Nevertheless, we find that infection in the Africa WHO region followed a "natural course", with episodes of infection acceleration aligned with the emergence of new variants, and death acceleration lagging by several weeks. The spikes in mortality closely mirrored earlier spikes in cases, almost one for one, reinforcing the pattern observed across the pandemic timeline. Given that our African subsample ranks among countries with relatively higher living standards, the overall pattern that we report in this paper should raise urgent concern among international and local health authorities, and it underscores the need for enhanced preparedness for future pandemics.

More broadly, this analysis has focused on the informational value of tests from surveillance systems. However, tests have an additional role in pandemics: they can drive real-time changes in behavior through individual decisions and policy interventions, such as limiting contacts or promoting isolation, making them a critical tool in shaping responses to future pandemics (see ref. 24). Incorporating "diagnostic effort" into modern epidemiological frameworks is, however, crucial for accurately tracking and understanding the spread of infectious diseases like COVID-19. Although we demonstrated this through the use of acceleration indices, other approaches should also be explored. In our view, this opens new and promising avenues for real-time pandemic management, potentially enhanced by AI algorithms that integrate "diagnostic effort" and its allocation across groups and space into dashboards of surveillance systems.

## Data availability
The harmonized databases used in this study are available in CSV format at https://github.com/PatrickPintus/COVID-in-WHO-regions, with no accession code required. Users are encouraged to cite the original data sources from which the material was collected. We additionally request that

any use of the newly constructed databases cite this study accordingly, accessible at https://doi.org/10.5281/zenodo.17305922 (see ref. 25). Source data for the box-plot Figs. 1–3 can be found in Supplementary Tables 2–4. Country data points correspond to simple yearly averages from the harmonized primary database cited above. Source data for Fig. 4 (infection and mortality raw and estimated acceleration indices, including estimated 95% CI bounds by WHO region) are also available at the same repository. Source data for Fig. 5 are provided as well, together with the R code required to generate the figure from the corresponding source data.

## Code availability
The R code, which delivers the acceleration index of infection (i.e., that of cases with respect to tests), is available at https://github.com/PatrickPintus/COVID-in-WHO-regions, without any accession code needed. It can be easily adapted to compute the acceleration index of mortality with respect to cases with access via https://doi.org/10.5281/zenodo.17305922. See ref. 25.

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

## Acknowledgments
This work was supported by the French National Research Agency Grant ANR-17-EURE-0020, and by the Excellence Initiative of Aix-Marseille University—A*MIDEX. M.S. acknowledges funding from the Agence Régionale de Santé de Nouvelle Aquitaine (ARS-SSMIP).

## Author contributions
S.L., C.P., P.P., M.S. and M.T. jointly conceived the study, designed the research framework, and developed the methodological approach. All authors contributed to data analysis, interpretation, and paper preparation. S.L., C.P., P.P., M.S. and M.T. reviewed and approved the final version of the paper.

## Competing interests
The authors declare no competing interests.
