## [Transparent Peer Review file · Communications Medicine]

A Comparative Analysis of Infection and Mortality in Reassessing Africa's COVID-19 Dynamic using Time-Varying Tests

Corresponding Author: Mr Stéphane Luchini

Version 0:

Reviewer comments:

Reviewer #1

(Remarks to the Author)

See attached file.

Reviewer #2

(Remarks to the Author)

I found the work is interesting and insightful. The idea of using acceleration overcomes the problem of under-reporting in less developed countries. My comments are minor:

I think the term “viral spread” may be replaced by “infection”, since “infection” and “mortality” are frequently used together in literatures.

There are other attempts to address the heterogeneous spread of COVID-19 in Africa, e.g., for the authors information.

<https://doi.org/10.1007/s11538-022-00992-x>

Reviewer #3

(Remarks to the Author)

The article presents an interesting analysis of COVID-19 cases, deaths, and testing dynamics across different regions worldwide. While I did not find inconsistencies in the methodology, I do not believe the study presents sufficiently novel findings to justify publication. In particular:

- Figures 1 to 3 report cases, tests, and deaths per capita in 2020 and 2021. However, very similar information is readily available on the Our World in Data dashboard, which serves as the study's primary data source. The authors' original contribution appears to be limited to minimal data preprocessing and cleaning. Consequently, the visualizations do not add substantial value beyond what is already publicly accessible and interactive online. The patterns emerging from Figures 1–3 are expected, particularly regarding testing capacity and cross-country heterogeneity. Some results also appear quite raw—for instance, case-fatality rates are not adjusted for age, making meaningful cross-country comparisons impossible.

- The analysis of the acceleration rate is more interesting, but I do not think it adds substantial insight. In particular, I do not find it to be strong support for one of the paper's key claims: that Africa was not spared by COVID-19 in 2020–2021. While I agree with this statement, I do not believe the acceleration index analysis supports it. My understanding is that the acceleration index captures only relative changes and does not provide information on absolute burdens, which would be necessary to robustly support the claim that Africa was indeed not spared.

- A minor but relevant point: it is unclear how the confidence intervals of the acceleration index are derived. If, as I understand, they are obtained from data across different countries, the paper may be incorrectly attributing wide CIs to measurement inaccuracy when they are more likely explained by underlying heterogeneity between countries.

Version 1:

Reviewer comments:

Reviewer #1

(Remarks to the Author)

Thank you for addressing my comments and suggestions. I only have a few remaining points.

1. page 7. add subscripts onto the fraction p/d to be consistent with the previous text.
2. page 7. Suggest changing the language where you describe the acceleration index. It is very close to the language in the Methods and Materials section of <https://journals.plos.org/plosone/article?id=10.1371/journal.pone.0281943> where in the index is described.
3. page 7. There is some confusion between daily/weekly rates in the text. Specifically, the left-hand side of equation 2 is referred to as both "weekly positivity rate" and "daily positivity rate".

Version 2:

Reviewer comments:

Reviewer #1

(Remarks to the Author)

I thank the authors for their hard work and improved manuscript. The central hypothesis of the manuscript, that Africa was not spared the worst of the COVID-19 pandemic in 2020 and 2021, while true, might be ascertained by considering the data in figures 1-3. Specifically, that Africa had significantly lower testing rates, but a higher positivity rate of those tests, and that the deaths as a proportion of cases was higher.

The newly added material in the appendix, requires further revision and technical clarification. Firstly, please define π and δ . In the if and only if statement in the first sentence of the second paragraph, the necessary condition is always satisfied. The derivative of $\log(f(x))$ is $1/f(x) f'(x)$ for all $x > 0$. Thus the ratio P/D is always increasing with time? This ratio is not monotonically increasing, as P and D are cumulative counts, and may be constant for some time period, e.g., when no new positive tests are identified.

In the last sentence of the second paragraph you write, "the derivative of the log of $P_i(t)/D_i(t)$ equals zero when...". This cannot be: $d \log(x)/dx$ is only defined for $x > 0$ and is undefined for $x = 0$. The gradient of the log is monotonically increasing, and therefore cannot be used to determine if the cumulative positivity rate is going to change direction.

In the last sentence of the third paragraph, you write, "this suggests that although the sign of the derivative of the log signals qualitatively whether the pandemic improves or worsens, its value is not a proper quantitative measure of how the pandemic evolves within and across groups." The sign is always positive, so the pandemic always worsens? We know this not to be true; please reconsider your analysis.

Lastly, the analysis presented in the appendix assumes that neither π nor $\delta = 0$ for all t . I'm not sure how these quantities are defined, but these singularities need to be addressed. A similar issue needs to be addressed with $D_T - D_{\{T-1\}}$ in equation 1 and with $P_T - P_{\{T-1\}}$ in equation 4. If there are no new diagnosed cases between T and $T-1$, then the denominator in equation 1 causes ϵ to be undefined. Similarly for ν and P_T and $P_{\{T-1\}}$ in equation 4.

From figure 4, from the rebuttal, the author's main points are: "First, the African Region experienced infection acceleration episodes of similar magnitude—but greater frequency—relative to other WHO regions. Second, uniquely among regions, these infection-acceleration episodes were consistently followed, several weeks later, by mortality-acceleration episodes of comparable magnitude. Given that our African subsample ranks among countries with relatively higher living standards, this pattern should raise urgent concern among international and local health authorities and underscores the need for enhanced preparedness for future pandemics."

The final point that the authors make is salient and should be emphasized within the manuscript. It appears that the WPR and SEAR experienced similar disease and mortality dynamics as the AFR, ie, infection-acceleration episodes followed by mortality-acceleration, so the claim uniqueness is perhaps overstated.

Also, the confidence intervals are negative in some places in figure 4; given the definition of these indices, the values should be always non-negative.

Reviewer #3

(Remarks to the Author)

I would like to thank the authors for their efforts in clarifying key methodological concepts, which effectively addressed my

previous concerns regarding the novelty and relevance of the approach and the findings.

I also appreciate the revisions made to the introductory section of the results, which help set the stage for the acceleration index insights. However, I still find this part somewhat overemphasized. In my view, the harmonization efforts described by the authors mainly reflect standard data cleaning procedures that are typically required in any project involving real world data. That said, this remains a matter of personal interpretation and does not represent a barrier to the acceptance of this article.

Version 3:

Reviewer comments:

Reviewer #1

(Remarks to the Author)

I thank the authors for their work in clarifying their arguments in the main text and appendix; the manuscript is much improved. I have no further comments.

Reassessing Africa's COVID-19 Dynamics: A Comparative Analysis of Infection and Mortality Across WHO Regions Taking Into Account Time-Varying Tests

point-by-point response to the referees' comments (highlighted in blue)

Reviewer #1 (Remarks to the Author):

Summary

The manuscript describes an analysis of the progression of Covid-19 in Africa, specifically looking at the African “paradox”, i.e., the narrative that Africa was spared the worst of the pandemic due to demographic and other factors. The study introduces two different ways to quantify the disease burden and how the rate of infection or mortality is changing in time. These indices are well motivated and the text contains multiple analyses showing a comparison between the numbers of infections and mortalities in the African region and five other regions as defined by the World Health Organization. As studied through the statistical analysis provided by the authors, the African “paradox” appears to vanish.

We sincerely thank the reviewer for their thoughtful summary of our work. We appreciate their recognition of our approach in quantifying the disease burden and analyzing the evolution of infection and mortality rates. Their comments reinforce the importance of our analysis in reassessing the African "paradox," and we are grateful for their constructive feedback. Below, we provide a point-by-point response to their comments and questions.

Overall Comments/Questions

1. What may be a nice analysis of health discrepancies in different regions is unfortunately marred by inconsistent writing and unclear/insufficient justification. Primarily, the data description needs to be addressed. It is not clear to this reader how the difference between number of cases and number of tests administered could produce a ratio in equation 1 which is greater than 1. The only way to

determine a positive case is through testing, hence the number of tests in each week should be greater, or equal to, the number of positive cases. I suspect this is related to some of the preprocessing work described in sect 2.1.1; it is not clear if the data used in the experiments is on a daily or weekly time scale.

Equation (1) provides the definition of the acceleration index, which is essentially a ratio of growth rates: that of **cumulated** positive cases divided by that of **cumulated** tests, over the considered period. We agree with the referee that the number of positive cases in any given period has to be smaller than or equal to the number of tests performed over that period. This, however, does not preclude the acceleration index from being smaller or larger than one: in the former case, we say that there is deceleration (the pandemic improves), while there is acceleration (the pandemic worsens) in the latter case. In other words, acceleration of the pandemic means that the growth rate of cumulated cases is larger than that of cumulated tests.

This clarification has been added to the revised version (highlighted in blue on p. 7, Section 2.2.2).

All results reported in the analysis of the dynamics are based on weekly data, as stated in the introduction and in Section 2.1, which provides a detailed and revised description of the database construction (highlighted in blue, Section 2.1, p.3).

2. What is the significance of the curves in figure 4? It appears that the estimated curves, from the acceleration indices, follow the raw data curves well, so I'm unsure what additional information is gained by plotting the acceleration curves as opposed to the raw data?

Figure 4 presents non-parametric estimations of acceleration indices for both infection and mortality, along with their corresponding confidence intervals. These estimations provide two key insights. First, the confidence intervals depicted in Figure 4 indicate whether we can reliably interpret the raw acceleration indices. Second, and most importantly for our analysis, the estimated confidence intervals enable the identification of statistically significant acceleration episodes in infection and mortality (i.e., acceleration indices exceeding 1), which are of critical importance in this study. We have now clarified the contribution of these non-parametric estimations in the revised version.

We have added an explanation for why we provided estimations and their corresponding confidence intervals in the revised version (highlighted in blue, p. 13, Subsection 3.2).

3. In section 3.3, description of which time-lags are used in the figure and analysis are not provided. It seems natural that there is a window of both positive and negative correlation between an increase in positive cases and disease mortality.

Figure 5 in this section is also very hard to read and interpret, due to a lot of information in limited space.

The description of each plot was provided in the note below the figure title to avoid overloading the main text. However, we understand the referee's concern regarding the amount of information presented. Despite its complexity, this figure is crucial as it clearly illustrates the coupling between viral spread and mortality episodes without relying on complex modeling techniques, which would require reductive assumptions and obscure direct observations of this relationship.

In the revised version, we have added new labels to the figure to enhance readability. Additionally, we have included a more detailed explanation of the figure in the main text (highlighted in blue in section 3.3, p.17).

4. The manuscript would benefit from careful editing and proof-reading. There are inconsistencies with naming, e.g., v_T vs. η_T , incomplete sentences, and typographic errors.

We have carefully proofread and edited the manuscript to correct inconsistencies in naming (e.g., v_T vs η_T), incomplete sentences, and typographic errors.

Reviewer #2 (Remarks to the Author):

I found the work is interesting and insightful. The idea of using acceleration overcomes the problem of under-reporting in less developed countries. My comments are minor:

I think the term “viral spread” may be replaced by “infection”, since “infection” and “mortality” are frequently used together in literatures.

There are other attempts to address the heterogeneous spread of COVID-19 in Africa, e.g., for the authors information.

<https://doi.org/10.1007/s11538-022-00992-x>

We have replaced the term “viral spread” by “infection” in the title and in the main text. Regarding the paper reporting heterogeneity across African countries that the referee alludes to, we have added a reference to it on page 3 of the Introduction section (highlighted in blue).

Reassessing Africa's COVID-19 Dynamics: A Comparative Analysis of Infection and Mortality Across WHO Regions Taking Into Account Time-Varying Tests

point-by-point response to the referees' comments (highlighted in red)

Reviewers' comments:

Reviewer #1 (Remarks to the Author):

Thank you for addressing my comments and suggestions. I only have a few remaining points.

1. page 7. add subscripts onto the fraction p/d to be consistent with the previous text.

Reply: Subscripts have been added in the main text, as requested.

2. page 7. Suggest changing the language where you describe the acceleration index. It is very close to the language in the Methods and Materials section

of <https://journals.plos.org/plosone/article?id=10.1371/journal.pone.0281943> where in the index is described.

Reply: A new, self-contained description of the acceleration index has been added in the main text, as requested, together with a new mathematical derivation in the Appendix.

3. page 7. There is some confusion between daily/weekly rates in the text. Specifically, the left-hand side of equation 2 is referred to as both "weekly positivity rate" and "daily positivity rate".

Reply: The requested changes have been made in the main text.

Reviewer #3 (Remarks to the Author):

The article presents an interesting analysis of COVID-19 cases, deaths, and testing dynamics across different regions worldwide. While I did not find inconsistencies in the methodology, I do not believe the study presents sufficiently novel findings to justify publication. In particular:

- Figures 1 to 3 report cases, tests, and deaths per capita in 2020 and 2021. However, very similar information is readily available on the Our World in Data dashboard, which serves as the study's primary data source. The authors' original contribution appears to be limited to minimal data preprocessing and cleaning. Consequently, the visualizations do not add substantial value beyond what is already publicly accessible and interactive online. The patterns emerging from Figures 1–3 are expected, particularly regarding testing capacity and cross-country heterogeneity. Some results also appear quite raw—for instance, case-fatality rates are not adjusted for age, making meaningful cross-country comparisons impossible.

Reply: Thank you for this constructive feedback. While the Our World in Data dashboard provides valuable country-level plots, our work went well beyond minimal "cleaning." We carried out extensive harmonization of reporting inconsistencies to build a coherent WHO-region-level database (see Materials & Methods, first revision in blue), and then used it to generate aggregated visualizations of cumulative tests, cases, positivity rates and case-fatality rates that are not directly accessible online. These regional plots are essential groundwork: they allow readers to assess data quality and contextualize the subsequent acceleration-index analysis.

Importantly, cumulative tests, cases, positivity rates and case-fatality rates are the very inputs for computing our acceleration indices. By first examining these yearly data at the WHO-region level, we ensure that the acceleration index reflects true epidemic dynamics rather than data artifacts. As the referee says, “these results are expected” and this comforts why they are relevant to the readers. This step is therefore indispensable for demonstrating, as we do, that the African Region did indeed experience statistically significant acceleration episodes in both infection and mortality.

With respect to age adjustment, our primary aim was to test the claim that Africa was “spared” during the COVID-19 pandemic, not to exhaustively model all drivers of cross-country differences. Although detailed age-standardization and causal analyses to unpack the drivers of cross-country and cross-region differences in pandemic dynamics would undoubtedly enrich the narrative, they lie beyond this paper’s scope. Nevertheless, in ancillary work we calculated correlations between cases per test and structural variables (temperature, median age, proportion ≥ 65 years, urbanization rate, malaria prevalence). While younger, less urbanized, malaria-endemic countries initially appeared to report fewer cases, those associations disappeared when normalizing by testing volume—and graphical inspection revealed no clear trend. We plan to explore those explanatory factors in a dedicated follow-up study.

We hope this clarifies both the originality and necessity of our preprocessing, regional visualizations, and analytical scope. Thank you again for prompting us to make these points more explicit.

- The analysis of the acceleration rate is more interesting, but I do not think it adds substantial insight. In particular, I do not find it to be strong support for one of the paper’s key claims: that Africa was not spared by COVID-19 in 2020–2021. While I agree with this statement, I do not believe the acceleration index analysis supports it. My understanding is that the acceleration index captures only relative changes and does not provide information on absolute burdens, which would be necessary to robustly support the claim that Africa was indeed not spared.

Reply: Thank you for noting your agreement that Africa was not spared by COVID-19 in 2020–2021. We regret that the novelty and rigor of our acceleration-index methodology were not sufficiently clear, so we have added a new, self-contained description of the index in the main text along with a full mathematical derivation in the Appendix.

The starting point of our approach is the recognition that exact knowledge of absolute burden—total infections and deaths—would require testing the entire population, which was neither feasible in general nor practised during COVID-19. Instead, one should adjust reported case counts by the number of tests performed. A common adjustment is the (daily or weekly) positivity rate: the ratio of new cases to tests in a given period, interpreted as a short-run infection speed. Likewise, the cumulative positivity rate—

cumulative cases over cumulative tests since the pandemic's start—serves as an average infection speed over the full time window.

However, because observed testing effort fluctuates over time (due to variations in capacity and willingness), positivity rates alone cannot track epidemic dynamics with precision. Our contribution is to extend the widely accepted reproduction-number concept to account for time-varying tests, yielding an acceleration index that is unit-free and computed on an open time window beginning with the first available data on tests, infections, and deaths. In the Appendix we show that this index captures turning points in the cumulative positivity rate for any group (e.g., region or demographic), allowing scale-free comparisons across groups and sub-periods. Crucially, when the acceleration index reaches zero, it signals a temporary halt in new cases—the cumulative burden plateaus—even if testing continues.

More generally, our goal is to provide a novel, real-time indicator of pandemic dynamics that informs both the public and health authorities whether the situation is improving or worsening. We believe that the greater the uncertainty about a pathogen, the more valuable such an index becomes. Moreover, it can enhance real-time surveillance systems and complement evidence from seroprevalence studies.

Finally, the claim that our subsample of African countries was not spared by COVID-19 rests on two key findings. First, the African Region experienced infection-acceleration episodes of similar magnitude—but greater frequency—relative to other WHO regions. Second, uniquely among regions, these infection-acceleration episodes were consistently followed, several weeks later, by mortality-acceleration episodes of comparable magnitude. Given that our African subsample ranks among countries with relatively higher living standards, this pattern should raise urgent concern among international and local health authorities and underscores the need for enhanced preparedness for future pandemics.

- A minor but relevant point: it is unclear how the confidence intervals of the acceleration index are derived. If, as I understand, they are obtained from data across different countries, the paper may be incorrectly attributing wide CIs to measurement inaccuracy when they are more likely explained by underlying heterogeneity between countries.

Reply: Thank you for your insightful comment. Our objective in this article is to study the dynamics of infection and mortality at the WHO-regional level by computing a single acceleration index for each region (one for infections and one for mortality), aggregating data from multiple countries. From a conceptual standpoint, this aggregation is valid; however, as you correctly observe, the resulting confidence intervals capture both statistical uncertainty and genuine heterogeneity in epidemic trajectories, data completeness, and public-health responses across countries—not merely measurement imprecision. We intentionally report these wide CIs to acknowledge that heterogeneity and to demonstrate that, despite

between-country variation, the African Region experienced a statistically significant acceleration in both cases and deaths.

In our first revision we expanded the Results section to address referee feedback on confidence intervals generally; in this second revision, we have now added an explicit statement (in red) in the Results section, clarifying that the principal driver of the interval width is cross-country heterogeneity and to discourage interpreting them as mere sampling error. Thank you again for helping us sharpen this point.

**Reassessing Africa's COVID-19 Dynamics: A Comparative Analysis of Infection and Mortality
Across WHO Regions Taking Into Account Time-Varying Tests**

point-by-point response to the referees' comments (highlighted in red)

Note: All additions appear in orange in this 3rd revision of the article.

Reviewers' comments:

Reviewer #1 (Remarks to the Author):

I thank the authors for their hard work and improved manuscript. The central hypothesis of the manuscript, that Africa was not spared the worst of the COVID-19 pandemic in 2020 and 2021, while true, might be ascertained by considering the data in figures 1-3. Specifically, that Africa had significantly lower testing rates, but a higher positivity rate of those tests, and that the deaths as a proportion of cases was higher.

Reply: We thank the referee for highlighting how Figures 1–3—by showing Africa's comparatively low testing rates alongside elevated test-positivity and case-fatality ratios over 2020 and 2021—already foreshadow the patterns we uncover in our weekly dynamic analysis. We have revised the manuscript (see page 13) to explicitly state that these annual snapshots both motivate and reinforce our dynamic investigation in the main text.

The newly added material in the appendix, requires further revision and technical clarification. Firstly, please define π and δ .

Reply: We have added on page 2 of the appendix a sentence to emphasize that the scale (i.e. level) parameters α and γ capture the fact that some groups may be more or less tested, or more or less susceptible to the virus, than others. In addition, the functions $\pi(t)$ and $\delta(t)$ capture that the dynamics of cases and tests over time may also be group-specific.

In the if and only if statement in the first sentence of the second paragraph, the necessary condition is always satisfied. The derivative of $\log(f(x))$ is $1/f(x) f'(x)$ for all $x > 0$. Thus the ratio P/D is always increasing with time? This ratio is not monotonically increasing, as P and D are cumulative counts, and may be constant for some time period, e.g., when no new positive tests are identified.

Reply: We thank the referee for helping us clarify this important point. We have clarified in the appendix (page 2) that the cumulated numbers of cases and tests P and D are nondecreasing, which does not imply that the ratio P/D is always increasing over time. This is why a condition is needed and is stated on page 2 of the Appendix. As explained in the revised version of the Appendix, the

condition is stated in terms of the log of the ratio P/D. This ensures that the analysis does not depend on the scale parameters alpha and gamma. Therefore, the discussion of how P/D varies over time is cast in terms of the derivative of $\log(P(t)/D(t))$ with respect to time.

In the last sentence of the second paragraph you write, "the derivative of the log of $P_i(t)/D_i(t)$ equals zero when...". This cannot be: $d \log(x)/dx$ is only defined for $x > 0$ and is undefined for $x = 0$. The gradient of the log is monotonically increasing, and therefore cannot be used to determine if the cumulative positivity rate is going to change direction.

Reply: As the discussion of how P/D varies over time is cast in terms of the derivative of $\log(P(t)/D(t))$ with respect to time, we have added in the text that this quantity is defined only if P and D are strictly positive. Since P and D are cumulated variables (i.e. stocks of cases and tests), they become strictly positive as soon as one test is carried out and one positive case is found, which is a necessary starting point if one is to study the spread of a disease. More generally, as soon as some diagnostic effort is spent to detect cases, such a condition is fulfilled. In addition, the number of new tests per week d cannot be zero. The latter condition is natural, as it essentially means that without new tests, there is no new information regarding the dynamics of the pandemic. As we have explained in the paper, the number of tests, not time, is the relevant unit of measurement (see page 7).

In the last sentence of the third paragraph, you write, "this suggests that although the sign of the derivative of the log signals qualitatively whether the pandemic improves or worsens, its value is not a proper quantitative measure of how the pandemic evolves within and across groups." The sign is always positive, so the pandemic always worsens? We know this not to be true; please reconsider your analysis.

Reply: As explained in the response to the preceding point raised by the referee, the derivative of the $\log(P(t)/D(t))$ can be negative, positive, or zero, even if $P(t)$ and $D(t)$ are non-decreasing over time. As stated on page 2 of the Appendix, $d \log(P_i(t)/D_i(t))/dt$ is positive if and only if $\pi'_i(t)/\pi_i(t) - \delta'_i(t)/\delta_i(t)$ is positive. When the growth rate of cases is larger than the growth rate of tests, P/D goes up and this indicates that the pandemic is worsening.

Lastly, the analysis presented in the appendix assumes that neither π nor $\delta = 0$ for all t . I'm not sure how these quantities are defined, but these singularities need to be addressed. A similar issue needs to be addressed with $D_T - D_{\{T-1\}}$ in equation 1 and with $P_T - P_{\{T-1\}}$ in equation 4. If there are no new diagnosed cases between T and $T-1$, then the denominator in equation 1 causes epsilon to be undefined. Similarly for ν and P_T and $P_{\{T-1\}}$ in equation 4.

Reply: We thank the referee for pointing this out. We have added new sentences on pages 7 and 8 to make clear that d cannot be zero, as discussed above. Similarly, p cannot be zero for ν to be defined. This is again a natural condition, since one cannot study either new cases without new tests

or new COVID 19-related deaths without new cases. As a consequence, there is no point in updating our acceleration indices when there is no new information. This is the very reason why we have spent extended time on building a consistent dataset through checking that there were no new cases without new tests and no new deaths without new cases. This point is emphasized on page 4 of Section 2.1.1 of the paper. The cleaned databases have been made available through an open web link provided in the paper.

From figure 4, from the rebuttal, the author's main points are: "First, the African Region experienced infection acceleration episodes of similar magnitude—but greater frequency—relative to other WHO regions. Second, uniquely among regions, these infection-acceleration episodes were consistently followed, several weeks later, by mortality-acceleration episodes of comparable magnitude. Given that our African subsample ranks among countries with relatively higher living standards, this pattern should raise urgent concern among international and local health authorities and underscores the need for enhanced preparedness for future pandemics."

The final point that the authors make is salient and should be emphasized within the manuscript. It appears that the WPR and SEAR experienced similar disease and mortality dynamics as the AFR, ie, infection-acceleration episodes followed by mortality-acceleration, so the claim uniqueness is perhaps overstated.

Reply: We thank the reviewer for highlighting the importance of linking our findings to broader public health preparedness. In response, we have incorporated a sentence p. 22 into the Discussion section. We also thank the reviewer for pointing out that infection-acceleration followed by mortality-acceleration episodes also occur in the Western Pacific (WPR) and South-East Asia (SEAR) Regions. To clarify and contextualize this, we have added that the dynamics of the Africa (AFR) Region is not unique (page 17) and that the need for testing and health-system strengthening also extends to other regions (page 21).

Also, the confidence intervals are negative in some places in figure 4; given the definition of these indices, the values should be always non-negative.

Reply: We thank the reviewer for pointing out the issue with negative confidence-interval bounds. In the previous version of the manuscript, we did not impose a non-negativity constraint on the confidence intervals because our main interest lies in identifying episodes where the lower bound of the interval was significantly above 1, which would indicate a statistically significant acceleration episode. In this context, negative values of the lower bound are not meaningful, as the index being estimated is strictly positive by definition, and such cases would simply imply the absence of a significant acceleration episode. However, we acknowledge that negative confidence intervals can be visually misleading or raise concerns about estimations. To address this, we have revised our estimation procedure. In the revised manuscript, we now estimate our smoothed confidence bands

on the log-scale—using the same span as before—and back-transform to the original metric. In practice, this means we fit a local polynomial regression (loess) to the logarithm of acceleration indices at the exact data points, compute 95 % intervals on the log scale, and exponentiate the resulting bounds. By construction, the lower bounds are now strictly positive, and Figure 4 has been updated accordingly (including its description; see page 15) as well as the Results section. This comes at a cost, as confidence intervals are now a little larger when acceleration indices are above 1, due to the exponential back transformation. However, all our significant results and conclusions remain.

Reviewer #3 (Remarks to the Author):

I would like to thank the authors for their efforts in clarifying key methodological concepts, which effectively addressed my previous concerns regarding the novelty and relevance of the approach and the findings.

I also appreciate the revisions made to the introductory section of the results, which help set the stage for the acceleration index insights. However, I still find this part somewhat overemphasized. In my view, the harmonization efforts described by the authors mainly reflect standard data cleaning procedures that are typically required in any project involving real world data. That said, this remains a matter of personal interpretation and does not represent a barrier to the acceptance of this article.

Reply: We thank the referee again for the constructive comments and suggestions. We acknowledge that data cleaning is a common preprocessing step. However, we also recognize that studies focusing solely on case counts may overlook inconsistencies arising from unadjusted, time-varying testing data. Because our acceleration-index methodology critically relies on the consistency of tests, cases and deaths both over time and relative to each other, generic cleaning rules risk leaving such inconsistencies undetected—hence our emphasis on detailed harmonization to raise awareness, not to overstate its novelty. To ensure transparency and reproducibility, we have made our cleaned, harmonized dataset publicly available, and the article explains the procedures used to build it. We have added a sentence on p. 4 of the revised version to highlight this point, and we now refer explicitly to the “harmonized” dataset in the Data Availability section (p. 22) to make this clearer..

Finally, Reviewer 1 pointed out that the annual summary statistics presented in Figures 1–3—showing lower testing, higher positivity, and elevated case-fatality ratios in Africa—already provide a meaningful and reassuring first check on our central hypothesis that Africa was not spared the worst of the pandemic. We have highlighted this point more clearly in the revised manuscript (p.13).